# The Expression of Major Facilitator Superfamily Domain-Containing Protein2a (Mfsd2a) and Aquaporin 4 Is Altered in the Retinas of a 5xFAD Mouse Model of Alzheimer’s Disease

**DOI:** 10.3390/ijms241814092

**Published:** 2023-09-14

**Authors:** Irena Jovanovic Macura, Ana Zivanovic, Milka Perovic, Jelena Ciric, Tamara Major, Selma Kanazir, Sanja Ivkovic

**Affiliations:** 1Institute for Biological Research “Sinisa Stankovic”, National Institute of Republic of Serbia, University of Belgrade, 11060 Belgrade, Serbia; irena.macura@ibiss.bg.ac.rs (I.J.M.); milkap@ibiss.bg.ac.rs (M.P.); jelena.ciric@ibiss.bg.ac.rs (J.C.); selkan@ibiss.bg.ac.rs (S.K.); 2Vinca—Institute for Nuclear Sciences, National Institute of Republic of Serbia, University of Belgrade, 11351 Belgrade, Serbia; anazivanovicmb@gmail.com; 3Faculty of Pharmacy, University of Belgrade, 11221 Belgrade, Serbia; majtamara@gmail.com

**Keywords:** Mfsd2a, Aqp4, retina, Alzheimer’s disease, BRB, transcytosis, *srebp1-c*, glymphatic system

## Abstract

Cerebral amyloid angiopathy (CAA) is characterized by amyloid β (Aβ) accumulation in the blood vessels and is associated with cognitive impairment in Alzheimer’s disease (AD). The increased accumulation of Aβ is also present in the retinal blood vessels and a significant correlation between retinal and brain amyloid deposition was demonstrated in living patients and animal AD models. The Aβ accumulation in the retinal blood vessels can be the result of impaired transcytosis and/or the dysfunctional ocular glymphatic system in AD and during aging. We analyzed the changes in the mRNA and protein expression of major facilitator superfamily domain-containing protein2a (Mfsd2a), the major regulator of transcytosis, and of Aquaporin4 (Aqp4), the key player implicated in the functioning of the glymphatic system, in the retinas of 4- and 12-month-old WT and 5xFAD female mice. A strong decrease in the Mfsd2a mRNA and protein expression was observed in the 4 M and 12 M 5xFAD and 12 M WT retinas. The increase in the expression of *srebp1-c* could be at least partially responsible for the Mfsd2a decrease in the 4 M 5xFAD retinas. The decrease in the pericyte (CD13+) coverage of retinal blood vessels in the 4 M and 12 M 5xFAD retinas and in the 12 M WT retinas suggests that pericyte loss could be associated with the Mfsd2a downregulation in these experimental groups. The observed increase in Aqp4 expression in 4 M and 12 M 5xFAD and 12 M WT retinas accompanied by the decreased perivascular Aqp4 expression is indicative of the impaired glymphatic system. The findings in this study reveal the impaired Mfsd2a and Aqp4 expression and Aqp4 perivascular mislocalization in retinal blood vessels during physiological (WT) and pathological (5xFAD) aging, indicating their importance as putative targets for the development of new treatments that can improve the regulation of transcytosis or the function of the glymphatic system.

## 1. Introduction

Vascular dysfunction in the brain is recognized as an important factor that drives cognitive impairment in Alzheimer’s disease (AD) [1,2]. The neurosensory retina is closely connected to the brain and it is the only CNS part feasible for direct, repeated, and non-invasive examination. Moreover, it is suggested that vascular changes in the retina can predict cognitive decline [3].

Cerebral amyloid angiopathy (CAA) [4] and blood–brain barrier (BBB) breakdown [5,6] are important characteristics of AD pathology. CAA is found in over 85% of AD patients, characterized by pathological Aβ deposits on blood vessels and other vascular abnormalities, and it was proposed as a marker of clinical AD [7]. The development of CAA indicates reduced amyloid clearance from the brain parenchyma [8], impaired amyloid transport across the endothelium, and the failure of amyloid degradation [9,10]. In AD, BBB impairments, such as changes in the pericyte coverage [11], tight junction (TJ) damage [12], and endothelial cell death [13], are considered as one of the principal causes of cerebral amyloidosis due to their essential role in clearing abundant cerebral Aβ [14,15].

The impaired blood–retinal barrier (BRB) in AD patients was associated with increased retinal vascular amyloidosis [16,17]. Importantly, a significant association between retinal and brain amyloid deposition was demonstrated in living patients [18,19] and in animal AD models [20,21]. The insufficient Aβ clearance from the blood vessels is associated with impaired transcytosis and/or a dysfunctional glymphatic system. Transcytosis was suggested to be the major mechanism for BBB functioning and the suppression of transcytotic pathways in the CNS is necessary for its proper function [22,23]. Major facilitator superfamily domain-containing protein2a (Mfsd2a) is specifically required to regulate transcytosis in the CNS endothelial cells of the BBB [22,24,25] and the increase in transcytotic vesicles in endothelial cells in Mfsd2a^−/−^ mice was shown to be responsible for the BBB leakage. Mfsd2a is emerging lately as a potential therapeutic target and its overexpression in the eye was reported to decrease retinal neovascularization and vascular leakage in mouse models of retinopathy [26]. In addition, the manipulation of the Wnt pathway was shown to regulate BRB integrity through the regulation of Mfsd2a-mediated caveolar transcytosis [27]. Recent findings also showed that the virally induced overexpression of Mfsd2a was able to reverse learning and memory impairments [28]. Moreover, a significant reduction in Mfsd2a expression in the brain microvasculature was observed during aging (26% in 12-month-old mice and 29% in 24-month-old mice) [29]. These findings put Mfsd2a in the focus of research as the possibility that its upregulated expression can protect the integrity of BBB and BRB marks it as a potentially universal target during normal aging and in the case of different pathologies. However, studies regarding the changes in Mfsd2a expression in retinal blood vessels in AD patients and animal AD models are lacking.

Another mechanism implicated in Aβ accumulation is the dysfunctional glymphatic system [30] and its inability to sufficiently clear amyloid from blood vessels [31]. An ocular lymphatic drainage system was also identified in rodent models, which relies on an aquaporin-4 (Aqp4)-dependent pathway to clear fluid and metabolites [32]. Aqp4 is a member of the aquaporin family and a transmembrane channel protein that can mediate the transport of water and other small molecule solutes between cells [33]. The increased expression of Aqp4 in the brain is associated with advanced age [34], and was shown to be in correlation with the severity of CAA [35,36]. Additionally, the loss of Aqp4 polarization is considered a factor for reduced Aβ clearance [31,34] although it is still unclear whether the loss of polarization is a consequence or a cause of Aβ accumulation [37]. Recent findings demonstrated that the administration of 5-caffeoylquinic acid can normalize Aqp4 perivascular mislocalization and increase Aβ clearance in an APP/PS2 AD mouse model, which improved cognitive deficits and neuronal functions [38]. Despite the promising progress made in the research of glymphatic system and AQP4, further translational studies are needed to explore their alterations during AD pathology [39]. In addition, the changes in Aqp4 expression and polarization in AD retina during aging have not been examined so far.

In the present study, we used 5XFAD transgenic mice to assess the changes in the expression levels of Mfsd2a and Aqp4 as regulators of Aβ clearance from the blood vessels. 5XFAD mice co-overexpress five human familial AD gene mutations, and their pathology recapitulates numerous pathological features of AD [40], including the significant vascular changes observed at the parenchymal microvasculature and leptomeningeal or penetrating vessels [41]. Importantly, studies have reported a marked increase in Aβ42/Aβ40 in the retina of 5XFAD mice [42]. Moreover, concurrent alterations in retinal vascular physiology, anatomy, and Aβ protein levels in 5xFAD mice were suggested to correspond to previously reported findings in human AD [43]. Only female mice were used, since previous reports suggest that female 5xFAD mice exhibit a stronger response to treatments, changes in the environment, or genetic manipulations [44]. Considering that aging presents a major risk for AD development, we analyzed the changes in Mfsd2a and Aqp4 expression in the 4- and 12-month-old (4 M and 12 M) wild-type (WT) and 5xFAD animals. At 4 M, the mild cognitive changes are already present in 5xFAD mice but the retinas appear structurally normal. 12 M mice are considered middle-aged and, importantly, recent findings identified specific proteomic and genomic changes in the middle-aged adults as markers for increased dementia risk 25 years later [45]. Thus, comparative analyses at this age were used to define age-related changes in the retinal expression of Mfsd2a and Aqp4 versus the changes that occur due to the progression of AD.

## 2. Results

### 2.1. Amyloid β Accumulates in Retinal Blood Vessels in 4 M and 12 M 5xFAD and 12 M WT Mice

5xFAD mice are characterized by the excessive production of APP harboring three familial mutations [40] and Aβ accumulation was shown in the brain blood vessels in 5xFAD mice [41]. To qualitatively confirm Aβ accumulation in the retinal blood vessels in 5xFAD mice, we used anti-Aβ antibody-6E10 (Figure 1A–D). A strong colocalization of Aβ (depicted in red) was detected in retinal blood vessels (labeled with lectin and depicted in green) from 4 M and 12 M 5xFAD retinas (Figure 1B,D). At the same time, no 6E10 staining was observed in blood vessels from 4 M WT retinas (Figure 1B). However, a sporadic accumulation of Aβ was observed in blood vessels of 12 M WT retinas (Figure 1C).

### 2.2. Mfsd2a mRNA and Protein Expression Are Decreased in the 4 M and 12 M 5xFAD and 12 M WT Retinas

Mfsd2a has been recognized as an important regulator of BBB and BRB integrity and blood vessel transcytosis [27,46], and the main site of Mfsd2a expression in the retina is in endothelial cells [47]. The BRB integrity in AD patients and animal models was shown to be compromised due to the accumulation of Aβ in and around the blood vessels. We sought to understand how the accumulation of Aβ in the retinal blood vessels affects the expression of Mfsd2a mRNA and protein in 5xFAD mice at 4 M and 12 M. A significant decrease in the expression levels of Mfsd2a mRNA was observed in the 4 M 5xFAD (3.27 fold) and 12 M 5xFAD retinas (13.15 fold) (Figure 2A). A similar downregulation of *Mfsd2a* expression was observed in the 12 M WT retinas (3.35 fold) (Figure 2A). The effects of age and genotype on the downregulation of Mfsd2a mRNA expression were similar (F_age_ = 7.407, *p* = 0.0151; F_genotype_ = 7.201, *p* = 0.0163) while the interaction of these factors did not have an effect (F_agexgenotype_ = 1.913, *p* = 0.1856). Mfsd2a was suggested as a target for the sterol regulatory element-binding protein 1-c (Srebp1-c) [26] and we observed a significant increase in *srebp1-c* expression in the retinas of 4 M 5xFAD (1.84-fold increase, Figure 2B). However, its expression levels in 12 M WT and 12 M 5xFAD retinas were not significantly altered in comparison to 4 M WT retinas, suggesting the possibility that different mechanisms of Mfsd2a regulation were engaged in aging. In addition, we observed a strong decrease (45%) in *srebp1-c* expression in 12 M 5xFAD retinas when compared to 4 M 5xFAD retinas. Neither age or genotype exhibited a significant effect on the *srebp1-c* expression in contrast to the interaction of these two factors that had a strong effect (F_age×genotype_ = 9.426, *p* = 0.0078).

The changes in Mfsd2a mRNA expression were accompanied by significantly reduced protein expression in 4 M 5xFAD retinas (4.3-fold), 12 M 5xFAD retinas (2.54-fold), and 12 M WT retinas (1.78-fold), when compared to the control, 4 M WT retinas (Figure 2C). The effect of genotype on Mfsd2a expression was more significant than the effect of age (F_genotype_ = 35.1, *p* = 0.0004, F_age_ = 7.26, *p* = 0.0273). However, a significant effect of the interaction between age and genotype (F_age×genotype_ = 21.11, *p* = 0.0018) was also observed.

It was shown that the changes in the levels of Mfsd2a expression in blood vessels could be in correlation with the changes in the degree of pericyte coverage [24]. In addition, the loss of pericytes was reported both in aging [16] and in AD mouse models [19], including 5xFAD mice [48].

We analyzed the pericyte coverage on the retinal blood vessels in 4 M and 12 M 5xFAD and WT mice to establish if there is an association with the changes in Mfsd2a expression. Immunohistochemical staining using a pericyte marker (CD13, red) and an endothelial cell marker (Lectin, green) was performed on the retinas from all the experimental groups (Figure 3A). The quantification of CD13 blood vessel coverage (Figure 3B) revealed a significant reduction in pericyte coverage in 4 M 5xFAD retinas (37% decrease), 12 M 5xFAD retinas (42% decrease), and in 12 M WT retinas (37% decrease). Both genotype and age affected the reduction in CD13/blood vessel coverage but age exhibited a stronger influence (F_genotype_ = 35.56, *p* < 0.0001; F_age_ = 61.24, *p* < 0.0001). In addition, the interaction between age and genotype also had a strong effect on the loss of pericyte coverage (F_age×genotype_ = 66.30, *p* < 0.0001). Therefore, the loss of pericytes may be another factor responsible for the decreased Mfsd2a expression.

### 2.3. The Expression Levels of Genes Regulating Cholesterol Synthesis Are Altered in the 4 M and 12 M 5xFAD and 12 M WT Retinas, While the Expression Levels of Genes Regulating Cholesterol Transport Remain Unchanged

An altered expression of the genes regulating cholesterol synthesis was reported in the retina of Mfsd2a^−/−^ animals [26,46], suggesting a possible feedback mechanism. In addition, the lipid composition of endothelial cells was recognized as an important factor in the regulation of transcytosis and barrier permeability [22]. Therefore, we analyzed the expression levels of the genes regulating cholesterol synthesis: liver X receptor beta (*Lxrβ*), responsible for the integration of the pathways of cholesterol input and output [49], and endoplasmic reticulum-bound 3-hydroxy-3-methylglutaryl-coenzyme-A reductase (*Hmgcr*), the rate-limiting enzyme in cholesterol synthesis [50] in the retinas of 5xFAD and WT mice. The qRT-PCR analysis showed that the expression levels of *Lxrβ* in the retina were decreased in 5xFAD retinas at both 4 M and 12 M when compared to 4 M WT retinas (an 8.32- and 4.92-fold decrease, respectively) (Figure 4A). Although *Lxrβ* expression was decreased in both 12 M WT and 12 M 5xFAD retinas (1.77- and 4.92-fold, respectively, when compared to 4 M WT), the effects were more prominent in 12 M 5xFAD mice (a 2.77-fold decrease when compared with 12 M WT). The changes in *Lxrβ* expression were significantly affected by both factors (age and genotype) and by their interaction, but the strongest influence was genotype (F_genotype_ = 71.91, *p* < 0.0001), followed by the factor interaction effect (F_age×genotype _= 12.54, *p* = 0.0033), and then with the influence of age (F_age_ = 5.825, *p* = 0.0301). The changes in the expression levels of *Hmgcr* in 5xFAD retinas followed a similar pattern as the expression levels of *Lxrβ*. The strongest decrease was observed in 4 M 5xFAD retinas (a 2.04-fold decrease) (Figure 4B). However, aging induced the expression of *Hmgcr* in the 12 M WT retina (a 1.48-fold increase compared with the 4 M WT) but this increase was not observed in the 12 M 5xFAD (a 2.14-fold decrease compared to the 12 M WT; and a 30% decrease compared to the 4 M WT) (Figure 4B). A two-way ANOVA revealed that the factor of genotype exerted a stronger effect on *Hmgcr* expression (F_genotype_ = 50.58, *p* < 0.0001) than the factor of age (F_age_ = 14.46, *p* = 0.0022). However, the interaction of these factors was not significant (F_age__×genotype _= 2.510, *p* = 0.1371).

We next sought to understand if Aβ accumulation in the retinal blood vessels altered the expression of the genes regulating retinal cholesterol turnover. The cells recycle cholesterol through a very efficient apolipoprotein-dependent process. The most abundant is apolipoprotein E (ApoE) [51]. This process of lipidation is mediated by the ATP-binding cassette transporter A1 (ABCA1), which is located in plasma membranes and effluxes cholesterol and phospholipids out of many cells [51]. The qRT-PCR analyses showed that neither Abca mRNA nor ApoE mRNA expression were altered in any of the groups analyzed (Figure 4C,D).

### 2.4. Aqp4 mRNA and Protein Expression Are Increased in 4 M and 12 M 5xFAD and 12 M WT Retinas

As Aqp4 expression levels in the brains of 5xFAD mice were shown to increase with age and the progression of disease [34], we measured Aqp4 mRNA and protein expression in the 5xFAD retinas. Aqp4 mRNA were elevated in 4 M and 12 M 5xFAD retinas when compared to 4 M WT (a 3.77- and 4.71-fold increase, respectively) (Figure 5A). The aging also induced *Aqp4* expression and we observed a 3.22-fold increase in Aqp4 mRNA in the 12 M WT retinas when compared to the 4 M WT (Figure 5A). The effect of genotype on Aqp4 mRNA expression (F_genotype_ = 26.7, *p* < 0.0001) was stronger than the effect of age (F_age_ = 10.49, *p* = 0.0046). The interaction of these factors did not affect Aqp4 mRNA expression (F_age×genotype_ = 3.229, *p* = 0.0891). We have measured the retinal Aqp4 protein expression using immunohistochemistry (Figure 5B). The representative images showing the Aqp4 immunostaining in the retinas of 12 M WT and 4 M and 12 M 5xFAD are presented in Figure 5B. Increased protein expression levels of Aqp4 were observed in 4 M and 12 M 5xFAD retinas measured with ImageJ (a 1.93- and 1.94-fold increase, respectively), although the genotype factor did not have a significant effect on the Aqp4 expression (F_genotype_ = 1.665, *p* = 0.2213) (Figure 5C). Aging also affected Aqp4 expression, with an increase observed in the 12 M WT retinas when compared to the 4 M WT (a 2.42-fold increase; F_age_ = 16.58, *p* = 0.0015) (Figure 5C). The interaction of these factors also had a significant effect on Aqp4 expression (F_age×genotype_ = 15.86, *p* = 0.0018).

### 2.5. Perivascular Aqp4 Expression Is Decreased in 12 M Wild-Type and 5xFAD Retinas

Aqp4 expression is observed around the blood vessels and a disrupted perivascular AQP4 polarization caused by reactive astrogliosis was shown to impair glymphatic clearance in the models of AD and senescence [52]. In addition, reduced perivascular Aqp4 expression is significantly associated with increased Aβ burden [34]. We measured the changes in perivascular Aqp4-staining in the 4 M and 12 M WT and 5xFAD mice (Figure 6A,B) and observed a decrease in 12 M 5xFAD (19%) and 12 M WT (14%) retinas when compared to the 4 M WT. At the same time, there was no change in perivascular Aqp4 expression between the 4 M 5xFAD and 4 M WT retinas. The only effect on perivascular Aqp4 expression was the age factor (F_age_ = 21.15, *p* = 0.0006), while neither genotype nor the interaction of age and genotype had any effect (F_genotype_ = 7.642, *p* = 0.9932; F_age×genotype_ = 2.933, *p* = 0.1125).

## 3. Discussion

Retinal vascular Aβ deposits were reported in the retina of AD patients, and the postmortem AD retinas [3]. However, studies examining the changes in the expression of Mfsd2a and Aqp4, implicated in the regulation of Aβ clearance in AD retina, are lacking. In this study, we identified changes in the expression levels of Mfsd2a and Aqp4 in the retinas from 5xFAD female mice associated with the accumulation of Aβ in the blood vessels. In addition, we measured the perivascular localization of Aqp4 on the retinal blood vessels.

The observed that Aβ accumulation in the retinal blood vessels in 5xFAD mice is in agreement with the evidence of vascular amyloidosis in other AD mouse models (APPSWE/PS1ΔE9 and Tg2576 mouse) [21,53], and in the postmortem retinas of MCI and AD patients [3,18]. Furthermore, arterial narrowing, decreased capillary density [54], and other functional and structural changes were observed in the retinas of 5xFAD mice [55], although BRB preservation in 5xFAD retinas was also reported [56]. However, the changes in the regulation of transcytosis can impair blood barriers before any changes in the expression of tight junction proteins are observed [22]. Thus, the deregulation of Mfsd2a and the subsequent impaired transcytosis may obstruct Aβ clearance from the blood vessels. We observed a strong decrease in Mfsd2a expression, both at the mRNA and protein levels, in 4 M and 12 M 5XFAD retinas and 12 M WT retinas, indicative of a compromised BRB. One way of regulating Mfsd2a expression might be through the Srebp pathway, one of the highly increased signaling pathways in the eyes of Mfsd2a knockout mice [57]. Similarly, Mfsd2a downregulation in diabetic retinopathy was associated with the upregulation of Srebps [26]. We found that the expression of *srebp-1c* was upregulated simultaneously with the decrease in Mfsd2a expression in 4 M 5xFAD retinas. However, although the expression levels of Mfsd2a in aged 12 M retinas in both 5xFAD and WT were decreased in comparison to the 4 M WT retinas, the levels of *serbp1-c* were unaltered in these groups (Figure 2). It was shown that decreased Mfsd2a expression correlates directly with the reduced degree of pericyte coverage [24]. Therefore, endothelial–pericyte interactions may be the additional mechanism for the control of Mfsd2a expression which, in return, controls BRB integrity. Pericyte loss was reported as part of retinal blood vessel pathology both in AD and in aged mice [17,21]. The decreased expression of CD13, a pericyte marker, that we observed in 4 M and 12 M 5xFAD and 12 M WT retinas (Figure 3) may be responsible for the decreased Mfsd2a expression in both transgenic and aged retinas.

Recent studies showed that treatment with the Lxr agonist T0901317 induced an increase in Mfsd2a expression [57] and chromatin immunoprecipitation sequencing (ChIPseq) and gene array studies revealed Lxrβ binding sites in mouse Mfsd2a intron 3 [58]. Our results showed a decreased Lxrβ expression in the 5xFAD retinas, implicating another mechanism of Mfsd2a expression regulation. The downregulation of Lxrβ consequently decreased the expression of Hmgcr, the key factor in the regulation of cholesterol synthesis. Mfsd2a has been identified as a lipid transporter delivering docosahexaenoic acid (DHA) into the brain [59] and the lipid composition of CNS endothelial cells, particularly the cholesterol content, is proposed as a key factor in the regulation of transcytosis and barrier permeability [22]. The suggested mechanism proposes that the increased levels of DHA cause the displacement of cholesterol and Cav-1 in the membrane, thereby inhibiting caveolae formation and transcytosis (Figure 7A) [60]. Thus, the downregulation of genes regulating cholesterol synthesis in the 5xFAD retinas can be a compensatory mechanism aiming to inhibit transcytosis. Nevertheless, analyses of cholesterol content in the retinal endothelial cells in AD patients and AD mouse models are missing. In addition, deciphering the exact role of the proteins responsible for cholesterol synthesis in the regulation of Mfsd2a expression and transcytosis warrants further studies.

Another regulator of Aβ clearance, Aqp4, is expressed on glial cells that ensheathe the vasculature (astrocytic end feet in the brain and Muller glia cells in the retina) [61,62] and regulates the perivascular exchange of CSF and interstitial fluid. Aqp4 levels in the retina were shown to increase with age [34] and the loss of Aqp4 significantly increased amyloid deposition in the cerebral cortex [63] and exacerbated the deterioration of neuronal function in 5xFAD mice (Figure 7B) [64]. In addition, the inhibition of Aqp4 with the addition of a small molecule inhibitor, TGN-020, decreases Aβ40 drainage around the cerebral vessels [65]. These findings suggest the potential neuroprotective role of Aqp4. Therefore, the increased expression of Aqp4 that we observed in the retinas of both 4 M and 12 M 5xFAD, and 12 M WT mice, might be the compensatory mechanism induced by the increased Aβ accumulation in the retinal blood vessels.

The loss of perivascular Aqp4 localization associated with aging [52] and AD [34,61] suggests that the mislocalization of Aqp4 may slow glymphatic function and promote protein aggregation and neurodegeneration. The normalization of perivascular Aqp4 mislocalization, with the addition of 5-caffeoylquinic acid, led to increased Aβ clearance and improved cognitive impairment in an APP/PS2 AD mouse model [38]. Our findings revealed that perivascular Aqp4 expression is decreased in the 12 M 5xFAD and 12 M WT retinas, suggesting that the mislocalization of Aqp4 in 5xFAD retinas is associated with aging and possibly with the progression of AD pathology.

It has been postulated that Mfsd2a may serve to modulate transcytotic mechanisms in CNS endothelial cells for therapeutic purposes [66]. Recent research confirmed that storax attenuated BBB disruption by upregulating Mfsd2a and inhibiting Cav-1 in the endothelial cells in order to arrest the progression of cerebral ischemia [67]. Similarly, treatment with LPC-DHA, transported through Mfsd2a, strongly improved retinal function in 5xFAD mice [68], and treatment with fish oil significantly increased Mfsd2a expression and blood vessel coverage in the retinas of wild-type mice [69]. Finally, the overexpression of Mfsd2a in the eye was reported to decrease retinal neovascularization and vascular leakage in mouse models of retinopathy and, importantly, co-treatment with DHA had a synergistic positive effect [26].

Although Aqp4 plays a crucial role in the glial–lymphatic system, the precise mechanism with which Aqp4 affects Aβ accumulation in the blood vessels is not completely understood. On one hand, the deletion of Aqp4 was shown to reduce Aβ clearance and exacerbate Aβ peptide accumulation in the plaques and vessels of AD animal models [70]. On the other hand, the inhibition of Aqp4 function with TGN-020 significantly induced perivascular amyloid accumulation [65]. Besides the changes in the expression levels of Aqp4, altered perivascular localization is associated with the promotion of protein aggregation and subsequent neurodegeneration [34]. In addition, treatment with fish oil was shown to protect the Aqp4 polarization and to significantly improve Aβ clearance from the brain [71], implicating the potential for the targeted manipulation of the glymphatic system. Thus, AQP4 localization may become a useful target for therapies particularly designed to intervene with AD pathology prior to amyloid accumulation.

The limitations of the study and future research. This study was performed on the retinas from a 5xFAD AD mouse model [40], with the aggressive Aβ pathology that is much faster and more hostile compared to the time course of human sporadic AD. Therefore, it will be beneficial to examine Mfsd2a and Aqp4 expression changes in more humanized AD mouse models, such as APP knock-in, for example [72]. Considering that a significant correlation between retinal and brain amyloid deposition was observed in AD patients and animal models, further studies focusing on examining if there is a correlation between the changes in Mfsd2a and Aqp4 expression in brain versus retinal blood vessels are warranted.

## 4. Materials and Methods

### 4.1. Animals

All animal procedures complied with the Directive (2010/63/EU) on the protection of experimental animals or animals used for other scientific purposes. The Ethical Committee for the Use of Laboratory Animals approved these procedures (resolution No. 01–06/13, Institute for Biological Research, University of Belgrade, Belgrade, Serbia). Animal procedures used in this paper complied with the EEC Directive (86/609/EEC) on animal protection, including the efforts to minimize animal suffering. The animals were housed under standard conditions (12 h light/dark cycle, 23 ± 2 °C, relative humidity 60–70%), and their health status was routinely checked. Pelleted commercial rodent chow and water were available ad libitum.

5xFAD mice were purchased from the Jackson Laboratory (Cat. No: 3484-JAX and 100012-JAX, Bar Harbor, MA, USA) and a B6/SJL genetic background was maintained by crossing 5xFAD transgenic male mice with B6SJLF1/J female mice. Female F1-offspring were used in this study. 5xFAD mice overexpress mutant human amyloid precursor protein 695 with the Swedish mutation (K670N, M671L: elevates the production of total Aβ), Florida mutation (I716V: elevates the production of Aβ42, specifically), London mutation (V717I: elevates the production of Aβ42, specifically), and human presenilin 1 with 2 FAD mutations (M146L and L286V: elevates the production Aβ42, specifically) [40]. Non-transgenic, wild-type littermates (WT) were used as controls.

### 4.2. Tissue Collection

For the purpose of experimental analyses, animals were divided into four groups: 4-month-old WT (4 M WT), 4-month-old 5xFAD (4 M 5xFAD), 12-month-old WT (12 M WT), and 12-month-old 5xFAD (12 M 5xFAD) mice. Mice were anesthetized (100 mg/kg, Ketamidor, Richter Pharma, 16 mg/kg Xulased, Bioveta, Wells, Austria a.s. intraperitoneally), and each animal was perfused with 50 mL 0.1 M phosphate buffer (PBS) for 30 min and then decapitated. The eyes were enucleated, the optic nerve severed, and the cornea, lens, and vitreous body were removed. The eyecup, including retinal pigmented epithelium (RPE), choroid, and sclera, was separated and the retina was peeled off and removed for further analysis. All the tissue samples were stored at −80 °C prior to the RNA and protein isolation. Retinas from 7 eyes were processed separately for RNA isolation and qPCR. Western blot was performed on 6 retina samples. The eyes from WT and 5xFAD mice (n = 4) were enucleated and processed for immunohistochemistry.

### 4.3. Real-Time Quantitative Polymerase Chain Reaction (qRT-PCR)

#### 4.3.1. RNA Isolation, and Reverse Transcription

Total RNA was extracted from the eyes isolated from wild type and 5xFAD 4 M and 12 M animals (n = 7 per group) according to the manufacturer’s instructions for the TRIZOL isolation system (Invitrogen Life Technologies, Carlsbad, CA, USA). The RNA pellet was dissolved in 20 μL of DEPC water, RNA concentration was determined using spectrophotometry, and RNA integrity was verified by 1% agarose gel electrophoresis. Six μg of total RNA was treated with RNase-free DNase I (Thermo Fisher Scientific, Waltham, MA, USA) and reverse transcribed in the same tube with a High-Capacity cDNA Archive Kit (Applied Biosystems, Fister City, CA, USA), following the manufacturer’s protocol. The cDNA was stored at −20 °C until further use.

#### 4.3.2. Quantitative Real-Time RT-PCR (qRT-PCR)

20 ng of the resulting cDNA was used for PCR analysis in a final volume of 10 μL using RT2 SYBR Green qPCR Mastermix (Applied Biosystems, Fister City, CA, USA). RT-PCR amplifications were performed in an ABI 7500 thermal cycler (Applied Biosystems, Fister City, CA, USA) in the default cycling mode (50 °C for 30 min, 95 °C for 15 min, followed by 40 cycles of 94 °C for 60 s, 57 °C for 60 s, 72 °C for 60 s, and then incubation at 70 °C for 10 min). The results obtained using qRT-PCR were analyzed using RQ Study add-on v 1.1 software (Applied Biosystems/Thermo Fisher Scientific, Waltham, MA, USA), with a confidence level of 95% (*p* < 0.05). Quantification was performed using the 2^−ΔΔCt^ method [73] and the change in mRNA levels was expressed relative to the control value. Sequences of the used primers (by Vivogen, Belgrade, Serbia) are given in Table 1.

### 4.4. Western Blot Analysis

A total of 6 animals were used for each western blot analysis. Due to the small volume of retinal tissue, which is estimated to be ~10 mg when wet [74], and in order to improve the sensitivity of protein detection in the retina and the optic nerve, we pooled tissues from two retinas from different animals together [75]. In such a way, we were able to have all the groups analyzed presented on the same gel. Thus, each lane consisted of the 2 pooled retinas. In this way, there were three biological samples that were replicated two times [75]. The extracted retinal tissue was homogenized in 10 *w*/*v* of RIPA buffer as previously described [69]. Following centrifugation (21,000 rcf, 30 min, 4 °C), the supernatant was collected, and protein concentrations were determined using Micro BCA Protein Assay Kit (Pierce Biotechnology, Rockford, IL, USA). Equal amounts of proteins (15 μg) were loaded per lane and, after 10% SDS acrylamide gel electrophoresis separation, they were blotted onto nitrocellulose membranes (Amersham Bioscience, Piscataway, NJ, USA). Following incubation in the blocking solution (5% non-fat dry milk in Tris-buffered saline/0.1% Tween 20, TBST) at room temperature (RT) for 1 h, the membranes were incubated with rabbit polyclonal Mfsd2a (1:10,000, 10,539, Abcam, Cambridge, UK) primary antibody in TBST, overnight at +4 °C. The membranes were then incubated with the HRP labeled secondary anti-rabbit antibody (1:5000; sc-2370, Santa Cruz Biotechnology, Dallas, TX, USA) for 1 h in TBST at RT. The signal was detected (enhanced chemiluminescence, ECL, Amersham Bioscience, Piscataway, NJ, USA) following the exposure to an X-ray film. The signals were analyzed using computerized densitometry (image analysis program ImageQuant ver. 5.2, Amersham Bioscience, Piscataway, NJ, USA) with the Ponceau S staining of the membranes serving as the loading control. The relative values of the signals were normalized to the corresponding Ponceau S staining and statistically analyzed.

### 4.5. Immunohistochemistry

The eyes were enucleated and fixed in 4% paraformaldehyde at 4 °C (O/N), cryoprotected and embedded in 7.5% gelatin:15% sucrose. Then, 18 µm cryo-sections were used in the analysis. Sections were degelatinized at 37 °C for 30 min in 1xPBS and permeabilized using Triton (0.5%) for 15 min, followed by blocking (1% bovine serum albumin, BSA in 1xPBST (0.1% Triton in 1xPBS) for 1 h at RT, and incubation overnight at 4 °C with mouse monoclonal 6E10 (cat#803003, Biolegend, San Diego, CA, USA), rabbit polyclonal Aqp4 (1:500, AB3594, EMD Millipore, Burlington, MA, USA), and goat polyclonal CD-13 (1:100, AF2335, R&D Systems, Minneapolis, MN, USA) primary antibodies. To label brain vessels, sections were incubated with 488-conjugated Lycopersicon esculentum lectin (488 DL1174, Vector, Newark, CA, USA). Sections were subsequently washed in PBST and incubated with anti-rabbit, anti-mouse, and anti-goat secondary antibody conjugated to Alexa 568, Invitrogen, Carlsbad, CA, USA) used at a 1:500 concentration in PBS for 2 h at room temperature (RT). After washing in PBS, slides were covered with DAPI mounting medium (Merck, Darmstadt, Germany) and evaluated using fluorescent microscopy. Micrographs were captured on an Axio Observer Microscope (Z1 AxioVision 4.6 software system, Carl Zeiss, Germany) at a magnification of 20×. Findings reported by microscopic images were representative of observations performed in 3 separate stainings for each group (n = 4). In all pictures, the apical is up.

### 4.6. Quantification of Perivascular Aqp4 Expression

Quantification was performed from representative images taken on a Zeiss microscope (Zeiss, Baden-Württemberg, Germany) at 20× magnification with conditions kept identical for all groups. Aqp4- and the lectin-positive areas were measured using threshold processing (ImageJ software, NIH, Bethesda, MD, USA), and the perivascular Aqp4 expression was expressed as % area occupied by Aqp4 in the lectin-positive area. In each animal (n = 4 per group), 5 fields from 2 to 4 nonadjacent retinal sections were analyzed.

### 4.7. Statistical Analysis

Data were analyzed using Prism program (GraphPad Prism, Software, v.6, La Jolla, CA, USA). For multiple comparisons, two-way analysis of variance (ANOVA) (age and genotype as factors) with Tukey post hoc tests was performed. The test was considered significant when *p* < 0.05.

## 5. Conclusions

The findings in this study reveal the impaired Mfsd2a and Aqp4 expression and Aqp4 perivascular mislocalization in retinal blood vessels in terms of physiological (WT) and pathological (5xFAD) aging, indicating their importance as putative targets for new treatments and adjuvant therapies. Our analyses showed that Mfsd2a expression was significantly affected by genotype and that this effect was exacerbated with age. In contrast, age had the more significant effect on Aqp4 expression and vascular polarization. In addition, the expression of the genes regulating cholesterol synthesis (*Lxrβ* and *Hmgcr*) was predominantly affected by genotype. This is particularly important as the accumulation of Aβ in retinal blood vessels is the pathology common for three neurodegenerative-associated disorders related to aging that are, so far, without a cure—AD, glaucoma, and age-related macular degeneration (AMD) [76]—and the development of novel therapies regulating impaired Aβ clearance is warranted.

## Figures and Tables

**Figure 1 ijms-24-14092-f001:**
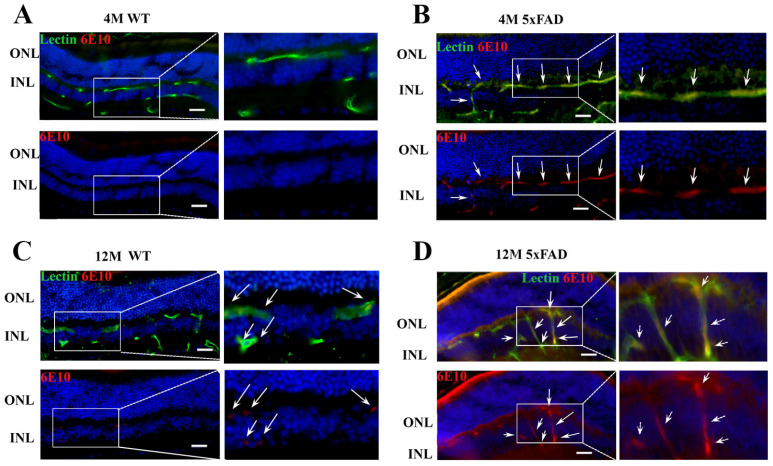
Aβ accumulation is present in the retinal blood vessels from 4 M and 12 M 5xFAD retinas and sporadically in the 12 M WT retinas. (**A**–**D**) The expression of Aβ in the retinal blood vessels from WT and 5xFAD mice was analyzed using immunohistochemistry. Co-localization was observed in the blood vessels from 4 M (**B**) and 12 M (**D**) 5xFAD retinas (arrows) and sporadically in the 12 M WT retinas (anti-Aβ, 6E10—red, lectin—green, DAPI—blue). Scale bar—50 μm. ONL—outer nuclear layer. INL—inner nuclear layer.

**Figure 2 ijms-24-14092-f002:**
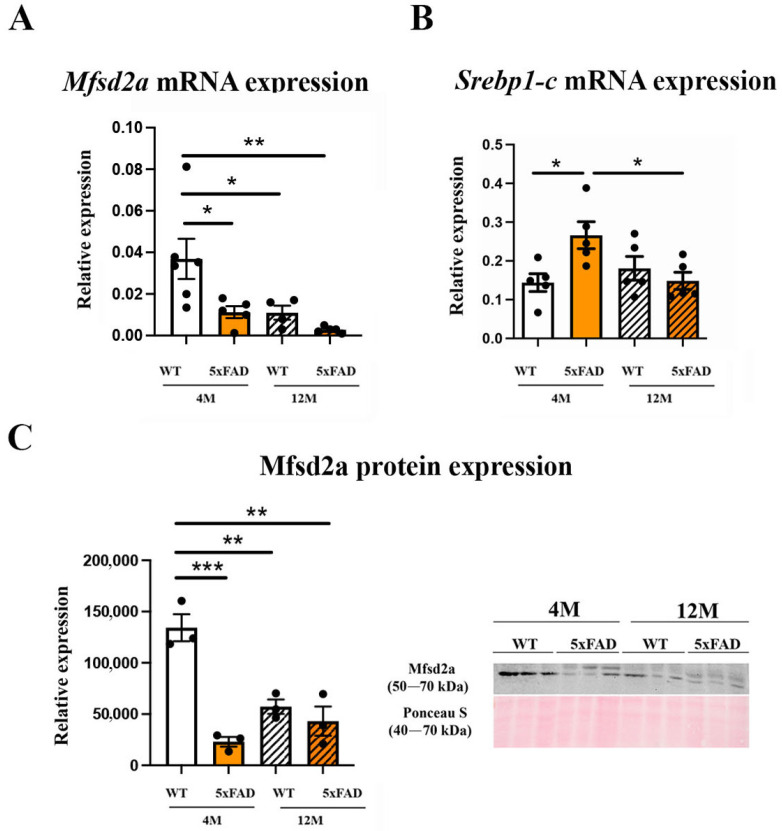
Mfsd2a expression is down-regulated in the retinas of 4 M and 12 M 5xFAD and 12 M WT mice. (**A**) The changes in the expression levels of *Mfsd2a* mRNA. (**B**) The changes in the expression levels of *Srebp1-c* mRNA. (**C**) The changes in the expression levels of Mfsd2a protein. Representative immunoblots of Mfsd2a in the 4 M and 12 M 5xFAD and WT retinas is presented. The data represent mean ± SEM value. * *p* < 0.05, ** *p* < 0.01, *** *p* < 0.0001.

**Figure 3 ijms-24-14092-f003:**
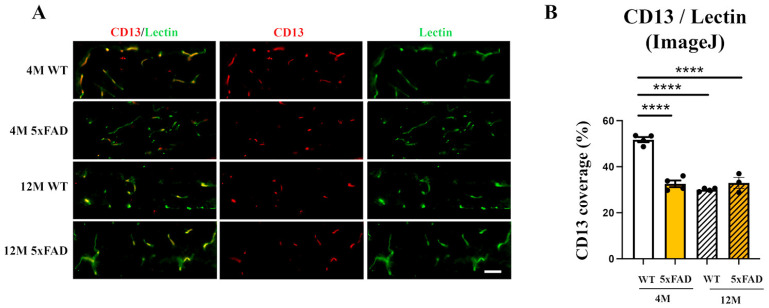
The pericyte coverage of retinal blood vessels is decreased in 4 M 5xFAD, 12 M WT, and 12 M 5xFAD mice. (**A**) Representative images of the degree of pericyte coverage (CD13, red) of retinal blood vessels (Lectin, green) in the 4 M WT, 4 M 5xFAD, 12 M WT, and 12 M retinas. (**B**) The quantification of the CD13 retinal blood vessel coverage was performed using ImageJ (National Institute of Health (NIH, USA)—Image J, Version 1.74) and the results were presented as the ratios of CD13/Lectin staining (graph). Scale bar—50 μm. The data represent mean ± SEM value. **** *p* < 0.0001.

**Figure 4 ijms-24-14092-f004:**
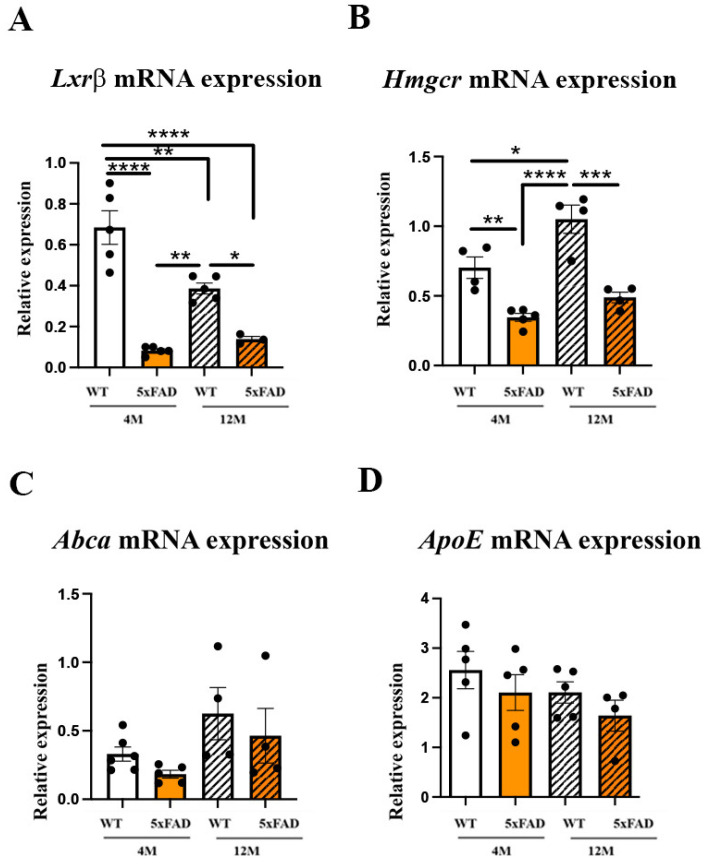
The expression levels of the genes regulating cholesterol synthesis are altered in the retinas of 4 M and 12 M 5xFAD and 12 M WT mice while the expression levels of the genes regulating cholesterol transport remain unchanged in all the groups analyzed. The expression levels of *Lxrβ* mRNA (**A**), *Hmgcr* mRNA (**B**), *Abca* (**C**), and *ApoE* (**D**) were analyzed in the retinas from 4 M and 12 M 5xFAD and WT mice using real-time polymerase chain reaction (qRT-PCR). Data are presented as mean ± SEM. * *p* < 0.05, ** *p* < 0.01, *** *p* < 0.001, **** *p* < 0.0001.

**Figure 5 ijms-24-14092-f005:**
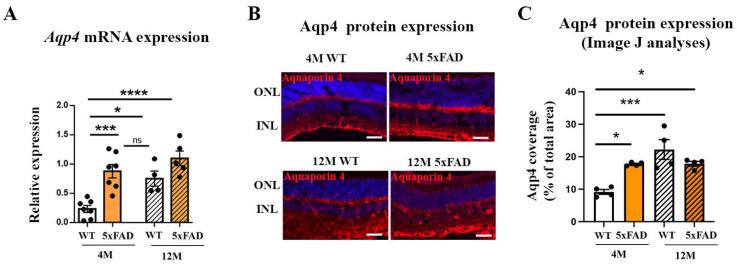
The expression levels of Aqp4 mRNA and protein were increased in the retinas of 4 M and 12 M 5xFAD and 12 M WT mice. (**A**) The changes in the expression levels of *Aqp4* mRNA. (**B**) The representative immunostaining of Aqp4 in the retinas of 4 M and 12 M, WT and 5xFAD mice. Aqp4—red, DAPI—blue. (**C**) The changes in the protein expression levels of Aqp4 measured with ImageJ analysis. Scale bar—50 μm. The data represent mean ± SEM value. * *p* < 0.05, *** *p* < 0.001, **** *p* < 0.0001. ns—not significant, ONL—outer nuclear layer, INL—inner nuclear layer.

**Figure 6 ijms-24-14092-f006:**
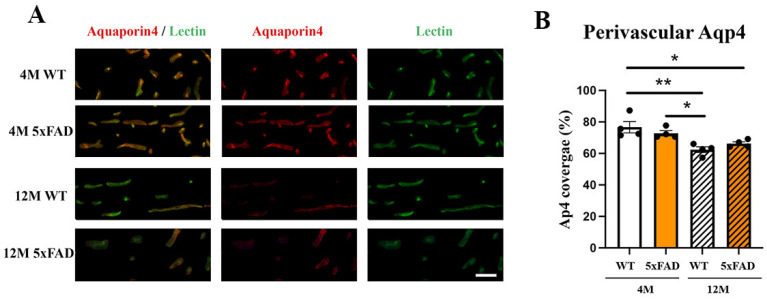
Perivascular Aqp4 expression is decreased in the 12 M WT and 5xFAD retinas. (**A**) Aqp4 perivascular expression in the 4 M and 12 M, WT and 5xFAD retinas was analyzed using immunohistochemistry (Aqp4—red, Lectin—green, DAPI—blue). (**B**) The quantification of the perivascular Aqp4 expression was performed using ImageJ and the results were presented as the ratio of Aqp4/Lectin staining (graph). Scale bar—50 μm. The data represent mean ± SEM value. * *p* < 0.05, ** *p* < 0.01.

**Figure 7 ijms-24-14092-f007:**
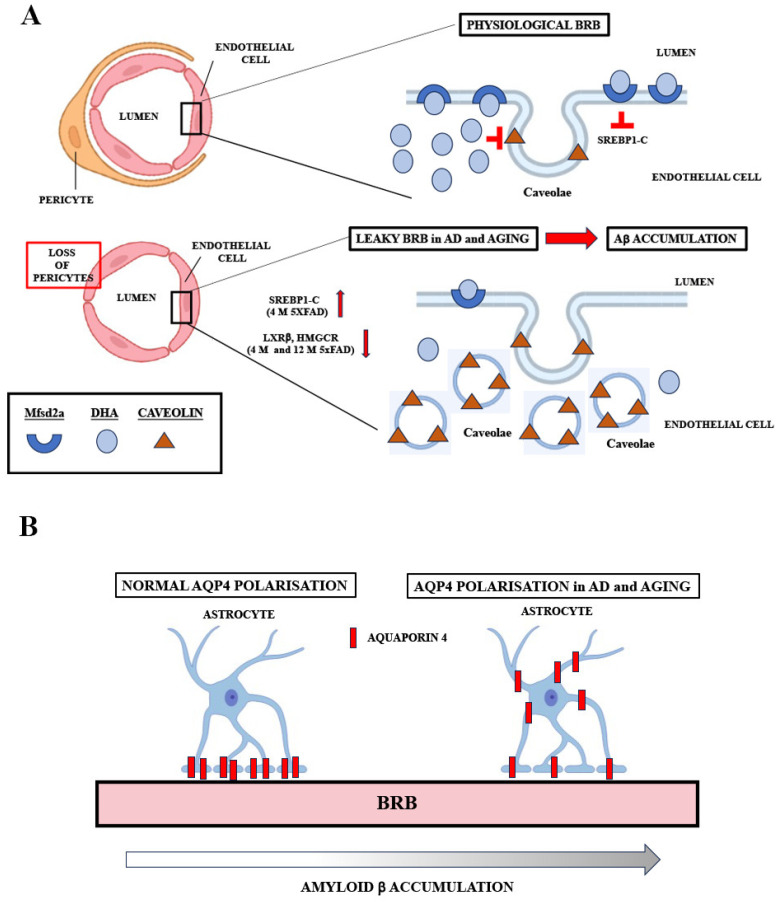
Schematic representations of potential mechanisms of Mfsd2a and Aqp4 actions at the BRB in AD and during aging. (**A**) In the normally functioning BRB, Mfsd2a-mediated DHA transport inhibits the formation of caveolae possibly through the inhibition of srebp1-c. In AD and aging, Mfsd2a downregulation, associated with the loss of pericytes, lessens the inhibition of the formation of caveolae, leading to increased vesicle transport through BRB, leaky barrier, and Aβ accumulation. In AD, the upregulation of srebp1-c and the downregulation of Lxrβ and Hmgcr could potentiate Mfsd2a downregulation. (**B**) Aqp4 expression is polarized, as it is densely expressed by astrocytes almost exclusively at the end feet, in direct contact with the perivascular space. In aging and in AD, Aap4 polarization is diminished. The loss of Aqp4 polarization is associated with increased Aβ accumulation.

**Table 1 ijms-24-14092-t001:** Primer sequences for expression studies.

Gene	Orientation	Sequence
*HMGCR*	F (5′-3′)	TTG GTC CTT GTT CAC GCT CAT
	R (3′-5′)	TTC GCC AGA CCC AAG GAA AC
*SREBP1-C*	F (5′-3′)	ACG GAG CCA TGG ATT GCA
	R (3′-5′)	AAG TCA CTG TCT TGG TTG TTGATGA
*LXRBETA*	F (5′-3′)	AGC GTC CAT TCA GAG CAA GTG
	R (3′-5′)	CAC TCG TGG ACA TCC CAG ATC T
*ABCA*	F (5′-3′)	AGG CCG CAC CAT TAT TTT GTC
	R (3′-5′)	GGC AAT TCT GTC CCC AAG GAT
*APOE*	F (5′-3′)	GGC CCA GGA GGA GAA TCA ATGA G
	R (3′-5′)	CCT GGC TGG ATA TGG ATG TTG
*MFSD2A*	F (5′-3′)	AGA AGC AGC AAC TGT CCA TTT
	R (3′-5′)	CTC GGC CCA CAA AAA GGA TAA T
*HPRT*	F (5′-3′)	CTC ATG GAC TGA TTA TGG ACA GGA C
	R (3′-5′)	GCA GGT CAG CAA AGA ACT TAT AGC C
*AQP4*	F (5′-3′)	AGC AAT TGG ATT TTC CGT TG
	R (3′-5′)	TGA GCT CCA CAT CAG GAC AG

F—forward primer, R—reverse primer.

## Data Availability

The data are contained within the article.

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
