# Peer review of "The Expression of Major Facilitator Superfamily Domain-Containing Protein2a (Mfsd2a) and Aquaporin 4 Is Altered in the Retinas of a 5xFAD Mouse Model of Alzheimer’s Disease"

_ijms, 2023, doi:10.3390/ijms241814092_

Round 1

Reviewer 1 Report

In this manuscript, Macura et al have investigated the changes in expression levels of Mfsd2a, the major regulator of transcytosis, and Aqp4, a key player of the lymphatic system in the retinas of 4- and 12-month old 5xFAD female mice. The results demonstrated altered expression of Mfsd2a and Aqp4 in the 5xFAD retinas highlighting these proteins as potential targets for the development of new therapeutics for Abeta clearance from blood vessels. Some of the results don't justify the conclusions and there are several other major concerns that need to be addressed for the manuscript to be considered for publication in IJMS. 

1. The abstract needs to be re-written with the focus on the most relevant results. Currently the abstract lacks some of the major highlights of the study. For example Msfd2a expression is altered in the 5xFAD retinas as early as 4 months of age. Msfd2a expression is also altered in 12-month old WT mice but these results suggest two different things.

2. The writing of the manuscript is confusing. It is not clear if the authors want to emphasize whether the changes in Msfd2a and Aqp4 levels are because of AD or because of aging or are both these factors important. The text should be modified accordingly.

3. The introduction  should focus more on the novelty of the research, the knowledge gap that currently exists, and how this study serves to fill the gap. In the introduction, some of the sentences are repetitive. For example "The retinal Abeta accumulation and vascular-associated deposits were reported in the retina of AD patients and patients with mild cognitive impairment (MCI), and the postmortem AD retinas [17,18].Retinal blood vessels in AD patients display morphological abnormalities such as disturbed blood flow dynamics, pericyte loss, and amyloid beta protein (Abeta) deposition [20]. 

4. It needs to be discussed in the manuscript why time points 4-months and 12-months were chosen.

5. "Amyloid beta is expressed in retinal blood vessels in 5xFAD mice". The term "expressed" is misleading since Amyloid beta is accumulated/deposited in retinal blood vessels. 

6. Figure 1. Include scale bar. Define ONL and INL in the figure legend. Abeta accumulation should be quantified. In the figure legend "A few Abeta+ plaques were observed in 12M 5xFAD retinas (C, arrowheads)". This sentence is incorrect.

7. Major amount of work is needed on the statistical comparisons for all data. The authors should also include in the methods section the information about the number of times the experiments were replicated.

8. Figure 2. Since the authors are focusing on the altered Msfd2a etc phenotypes in the 5xFAD mice, the statistical comparisons between WT and 5xFAD mice at both the time points should be included for all data.  How many animals were used for western blots? Currently, the blot indicates 3 animals for each group, while the relative expression graph has more data points. This difference should be properly explained.

CD13 blot should be replaced. Fig 2D results (blot and quantification) are not convincing. The Y-axis label in Relative expression graph in Fig 2D is missing and the X-axis labeling should be consistent with other graphs.

9. Figure 4. For Aqp4, the relative expression graph does not match the western blot because of which the results are not convincing. In 4C, Aquaporin 4 intensity should be quantified and better representative images should be included. Add scale bar.

10. Figure 5. Labels (A and B) and scale bars are missing. It appears from the images that 4-month old 5xFAD mice has reduced Aquaporin levels while the graph indicates no change in the levels. Please use a better representative image.

11. It is recommended that the authors include a schematic diagram to summarize how Mfsd2a, Aqp4 etc are regulated.

12. It is reported that in 5xFAD mice, Abeta accumulation starts as early as 1.5 months of age preceding amyloid deposition, gliosis and mitochondrial dysfunction at 2 months (Welding et al., 2019; Sharma et al., 2023). In this manuscript, the authors are trying to correlate Abeta clearance with Mfsda and Aqp4 levels. It is therefore crucial to include data from an earlier time point (preferable 2 months) to see if retinal accumulation of Abeta and changes in Mfsda and Aqp4 levels are coincidental. 

The paper is well written in sound English. However, there are certain typos that need to be corrected (Example Line 136-139 "Mfsd2a was 136 suggested as a target for the sterol regulatory element-binding protein 1-c (Srebp1-c) 137 [47] and we observed the significant increase of srebp1-c expression in retinas of 4M 138 5xFAD retinas (Figure 2B)".

Author Response

Reviewer 1.

In this manuscript, Macura et al have investigated the changes in expression levels of Mfsd2a, the major regulator of transcytosis, and Aqp4, a key player of the lymphatic system in the retinas of 4- and 12-month old 5xFAD female mice. The results demonstrated altered expression of Mfsd2a and Aqp4 in the 5xFAD retinas highlighting these proteins as potential targets for the development of new therapeutics for Abeta clearance from blood vessels. Some of the results don't justify the conclusions and there are several other major concerns that need to be addressed for the manuscript to be considered for publication in IJMS. 

  1. The abstract needs to be re-written with the focus on the most relevant results. Currently the abstract lacks some of the major highlights of the study. For example Msfd2a expression is altered in the 5xFAD retinas as early as 4 months of age. Msfd2a expression is also altered in 12-month old WT mice but these results suggest two different things.

We thank the reviewer for this comment. We have changed the Abstract according to the reviewer’s suggestions.

Abstract

Background Cerebral amyloid angiopathy (CAA) is characterized by amyloid b (Ab) accumulation in the blood vessels and is associated with the cognitive impairment in Alzheimer’s disease (AD).  The increased accumulation of Ab is also present in the retinal blood vessels and a significant correlation between retinal and brain amyloid deposition was demonstrated in living patients and animal AD models. Aim The Ab accumulation in the retinal blood vessels can be the result of impaired transcytosis and/or the dysfunctional ocular glymphatic system in AD and during aging. Methods We analyzed the changes in the mRNA and protein expression of major facilitator superfamily domain-containing protein2a (Mfsd2a), the major regulator of transcytosis, and of Aquaporin4 (Aqp4), the key player implicated in the functioning of the glymphatic system, in the retinas of 4-and 12-month-old WT and 5xFAD female mice. Results A strong decrease in the Mfsd2a mRNA and protein expression was observed in the 4 M and 12 M 5xFAD and 12M WT retinas. The increase in the expression of Srebp1-c could be at least partially responsible for the Mfsd2a decrease in the 4 M 5xFAD retinas. The decrease in the pericyte (CD13+) coverage of retinal blood vessels in the 4 M and 12 M 5xFAD retinas and in the 12 M WT retinas suggests that pericyte loss could be associated with the Mfsd2a downregulation in these experimental groups. The observed increase in Aqp4 expression in 4 M and 12 M 5xFAD and 12 M WT retinas accompanied by the decreased perivascular Aqp4 expression is indicative of the impaired glymphatic system. Conclusion The findings in this study reveal the impaired Mfsd2a and Aqp4 expression and Aqp4 perivascular mislocalization in retinal blood vessels during physiological (WT) and pathological (5xFAD) aging indicating their importance as putative targets for the development of new treatments that can improve the regulation of transcytosis or the function of glymphatic system.

  1. The writing of the manuscript is confusing. It is not clear if the authors want to emphasize whether the changes in Msfd2a and Aqp4 levels are because of AD or because of aging or are both these factors important. The text should be modified accordingly.

We thank the reviewer for this suggestion and we have incorporated the changes in the manuscript highlighted in yellow.

Aging is considered as a major risk for the development of AD and the changes in Mfsd2a and Aqp4 in the brain during physiological aging were reported. The decrease in Mfsd2a expression in the brain vasculature was evident starting at 12-months-old mice (26%) (Iwao et al., 2023). In contrast, the increase in Aqp4 expression is associated with aging and accompanied with the loss of Aqp4 polarisation (Zeppenfeld et al., 2017). However, the expression levels of Mfsd2a and Aqp4 were not investigated in the physiologically aged retina and in the aging AD retina so far.

Additionally, the accumulation of retinal Ab is suggested to be the common risk for other retinal diseases that are associated with aging such as age-related macular degeneration (AMD) and glaucoma (Wang and Mao, 2021). Thus, any information related to the expression changes of these two molecules in the retina during physiological aging that are implicated in the Ab clearance can be beneficial for the development of new targeted therapies.

Iwao T, Takata F, Matsumoto J, AridomeH, Yasunaga M, Yokoya M, et al. (2023) Agingdecreases docosahexaenoic acid transport acrossthe blood-brain barrier in C57BL/6J mice. PLoSONE 18(2): e0281946https://doi.org/10.1371/journal.pone.0281946

Zeppenfeld, D.M.; Simon, M.; Haswell, J.D.; D’Abreo, D.; Murchison, C.; Quinn, J.F.; Grafe, M.R.; Woltjer, R.L.; Kaye, J.; Iliff, J. J. Association of Perivascular Localization of Aquaporin-4 With Cognition and Alzheimer Disease in Aging Brains. JAMA Neurol.2017, 74, 91. doi:10.1001/jamaneurol.2016.437

Wang, L.; Mao, X. Role of Retinal Amyloid-β in Neurodegenerative Diseases: Overlapping Mechanisms and Emerging Clinical Applications. Int. J. Mol. Sci.2021, 22, 2360. doi: 10.3390/ijms22052360.

  1. The introduction  should focus more on the novelty of the research, the knowledge gap that currently exists, and how this study serves to fill the gap. In the introduction, some of the sentences are repetitive. For example "The retinal Abeta accumulation and vascular-associated deposits were reported in the retina of AD patients and patients with mild cognitive impairment (MCI), and the postmortem AD retinas [17,18]." Retinal blood vessels in AD patients display morphological abnormalities such as disturbed blood flow dynamics, pericyte loss, and amyloid beta protein (Abeta) deposition [20]. 

We have changed the introduction per reviewer’s suggestion. All the changes are highlighted yellow in the revised version of the manuscript.

  1. It needs to be discussed in the manuscript why time points 4-months and 12-months were chosen.

The changes in the retina are associated with the changes in the brain in AD animal models including 5xFAD mice (Zhang et al., 2021). However, the studies examining the changes in the retinas of 5xFAD mice are focused on different target molecules at different time-points. 5XFAD mice have reported a marked increase in Aβ42 (Pogue et al., 2015; Parthasarathy et al., 2015) at 3 months of age followed by some cognitive impairments as early as 4 months of age (Oakley et al., 2006). It is suggested that in the 5xFAD retinas structural changes preceded functional decay and at 4 months, retinas were generally normal (Zhang et al., 2021). Considering that the diagnosis of AD before the first onset of clinical symptoms is still not possible, we chose to evaluate the changes in the Mfsd2a and Aqp4 expression when the first signs of cognitive impairment present themselves.

After 6 months of age, the maturational rate of mice is 25 times faster than humans. Therefore, mice age 10-14 months are considered middle-aged. (https://www.jax.org/news-and-insights/jax-blog/2017/november/when-are-mice-considered-old). Recent study using proteomics and genomics on a cohort of middle-aged adults followed longitudinally, identified pathway-specific plasma proteins that increased dementia risk up to 25 years later (Walker et al., 2023). In the same time, the severity of pathological changes in the retinas in AD animal models was shown to increase with age (Habiba et al., 2020; Lim et al., 2021). Thus, the parallel analyses in the middle-aged group between 5xFAD and WT mice can help compare age-related changes versus the changes that occur due to the progression of AD.

We have added the appropriate explanations in the Introduction section of the manuscript.

Considering that the aging presents the major risk for the AD development we analyzed the changes in Mfsd2a and Aqp4 expression in the 4- and 12-month-old (4 M and 12 M) wild-type (WT) and 5xFAD animals. At 4 M, the mild cognitive changes already exist, however, the retinas in 5xFAD mice appear structurally normal. 12 M mice are considered middle-aged and, importantly, recent findings identified specific proteomic and genomic changes in the middle-age adults as markers for the increased dementia risk 25 years later (Walker et al., 2023). Thus, the comparative analyses at this age was used to define age-related changes in the retinal expression of Mfsd2a and Aqp4 versus the changes that occur due to the progression of AD.

Lim JKH, Li QX, Ryan T, Bedggood P, Metha A, Vingrys AJ, Bui BV, Nguyen CTO. Retinal hyperspectral imaging in the 5xFAD mouse model of Alzheimer's disease. Sci Rep. 2021 Mar 18;11(1):6387. doi: 10.1038/s41598-021-85554-2. PMID: 33737550; PMCID: PMC7973540.

Pogue, A.; Dua, P.; Hill, J.; Lukiw, W. Progressive inflammatory pathology in the retina of aluminum-fed 5xFAD transgenic mice. J. Inorg. Biochem. 2015, 152, 206–209. [CrossRef]

Parthasarathy, R.; Chow, K.M.; Derafshi, Z.; Fautsch, M.P.; Hetling, J.R.; Rodgers, D.W.; Hersh, L.B.; Pepperberg, D.R. Reductionof amyloid-beta levels in mouse eye tissues by intra-vitreally delivered neprilysin. Exp. Eye Res. 2015, 138, 134–144. [CrossRef]

Oakley, H.; Cole, S.L.; Logan, S.; Maus, E.; Shao, P.; Craft, J.; Guillozet-Bongaarts, A.; Ohno, M.; Disterhoft, J.; Van Eldik, L.; et al. Intraneuronal β-Amyloid Aggregates, Neurodegeneration, and Neuron Loss in Transgenic Mice with Five Familial Alzheimer’s

Disease Mutations: Potential Factors in Amyloid Plaque Formation. J. Neurosci. 2006, 26, 10129–10140. [CrossRef]

Habiba, U.; Merlin, S.; Lim, J.K.H.; Wong, V.H.Y.; Nguyen, C.T.O.; Morley, J.W.; Bui, B.V.; Tayebi, M. Age-Specific Retinal and Cerebral Immunodetection of Amyloid-β Plaques and Oligomers in a Rodent Model of Alzheimer's Disease. J. Alz. Dis.2020, 76, 1135-1150. doi: 10.3233/JAD-191346. PMID: 32597800

Zhang, M.; Zhong, L.; Han, X.; Xiong, G.; Xu, D.; Zhang, S.; Cheng, H.; Chiu, K.; Xu, Y. Brain and Retinal Abnormalities in the 5xFAD Mouse Model of Alzheimer's Disease at Early Stages. Front. Neurosci.2021, 15, 681831. doi: 10.3389/fnins.2021.681831

Walker KA, Chen J, Shi L, Yang Y, Fornage M, Zhou L, Schlosser P, Surapaneni A, Grams ME, Duggan MR, Peng Z, Gomez GT, Tin A, Hoogeveen RC, Sullivan KJ, Ganz P, Lindbohm JV, Kivimaki M, Nevado-Holgado AJ, Buckley N, Gottesman RF, Mosley TH, Boerwinkle E, Ballantyne CM, Coresh J. Proteomics analysis of plasma from middle-aged adults identifies protein markers of dementia risk in later life. Sci Transl Med. 2023 Jul 19;15(705):eadf5681. doi: 10.1126/scitranslmed.adf5681. Epub 2023 Jul 19. PMID: 37467317.

  1. "Amyloid betais expressed in retinal blood vessels in 5xFAD mice". The term "expressed" is misleading since Amyloid beta is accumulated/deposited in retinal blood vessels. 

We have changed the subtitle in the Result section per reviewer’s suggestion.

  1. Figure 1. Include scale bar. Define ONL and INL in the figure legend. Abeta accumulation should be quantified. In the figure legend "A few Abeta+ plaques were observed in 12M 5xFAD retinas (C, arrowheads)". This sentence is incorrect.

We thank the reviewer for this comment.

We have added the explanations for ONL and INL abbreviations in the legend of the Figure 1, the scale bar, and we omitted the sentence relating to the presence of the plaques.

Ab accumulation in the blood vessels of 5xFAD mice is somewhat controversial. In the study by Marazuela et al. (2022) no Aβ-positive vessels were detected in the brain from 16-month-old 5xFAD mice with the specific resorufin staining. On the other hand, Van Nostrand et al. (2002), reported CAA in leptomeningeal and penetrating brain vessels from 3- to 5-month-old 5xFAD mice by in vivo two-photon microscopy. To our knowledge there are very few papers showing Ab accumulation in 5xFAD retinal blood vessels so far including the recent study by Matei et al. (2022) showed Ab deposition in retinal blood vessels using rabbit anti-Aβ42 (1:100; Cell Signaling Technology Cat# 14974, RRID: AB_2798671). Thus, in order to confirm the Ab accumulation in retinal blood vessels, we performed immunohistochemical staining using mouse monoclonal 6E10 (cat#803003, Biolegend, San Diego, CA, USA) (Figure 1). As the results clearly showed the presence of Ab deposition in the retinal blood vessels in the 4 M and 12 M 5xFAD retinas and sporadically in the retinas from 12 M WT we concluded that there is amyloid accumulation in the retinal blood vessels in 5xFAD mice. Therefore, our goal was not to quantify the amount of Ab but just to show its presence in the retinal blood vessels in 5xFAD mice in order to confirm our hypothesis.

Marazuela, P.; Paez-Montserrat, B.; Bonaterra-Pastra, A.; Solé, M.; Hernández-Guillamon, M. Impact of Cerebral Amyloid Angiopathy in Two Transgenic Mouse Models of Cerebral β-Amyloidosis: A Neuropathological Study. Int. J. Mol. Sci.2022,23,4972. https://doi.org/ 10.3390/ijms23094972

Van Nostrand, W.E.; Melchor, J.P.; Romanov, G.; Zeigler, K.; Davis, J. Pathogenic Effects of Cerebral Amyloid Angiopathy Mutations in the Amyloid β-Protein Precursor. Ann. N. Y. Acad. Sci. 2002, 977, 258–265.

  1. Major amount of work is needed on the statistical comparisons for all data. The authors should also include in the methods section the information about the number of times the experiments were replicated.

All the data were reanalyzed using 2-wayANOVA with post hock Tuckey test. The effects of two factors age and genotype and the effects of their interactions are now included and presented for each quantification in the revised MS. The number of times the experiments were replicatedis also included in the MM section.

  1. Figure 2. Since the authors are focusing on the altered Msfd2a etc phenotypes in the 5xFAD mice, the statistical comparisons between WT and 5xFAD mice at both the time points should be included for all data.  How many animals were used for western blots? Currently, the blot indicates 3 animals for each group, while the relative expression graph has more data points. This difference should be properly explained.

The total of 6 animals were used for each western blot analysis. Due to the small volume of retinal tissue, which is estimated to be ∼10 mg when wet (Okawa et al., 2008) and to improve sensitivity of protein detection in the retina and the optic nerve, we pooled tissues from two retinas from different animals together (Smedowski, 2018). In such a way we were able to have all the groups analyzed presented on the same gel. Thus, each lane consists of the 2 pooled retinas. In this way, there were three biological samples that were replicated 3 times and quantified accordingly to previously reported studies with pooled samples. We have changed the graph in Figure 2C accordingly.

All the statistically significant changes are presented in the graphs including the changes, if there were any, between the WT and 5xFAD mice at both time points.

Okawa H, Sampath AP, Laughlin SB, Fain GL. ATP consumption by mammalian rod photoreceptors in darkness and in light. Curr Biol. 2008 Dec 23;18(24):1917-21. doi: 10.1016/j.cub.2008.10.029. Epub 2008 Dec 11. PMID: 19084410; PMCID: PMC2615811.

Smedowski A, Liu X, Podracka L, Akhtar S, Trzeciecka A, Pietrucha-Dutczak M, Lewin-Kowalik J, Urtti A, Ruponen M, Kaarniranta K, Varjosalo M, Amadio M. Increased intraocular pressure alters the cellular distribution of HuR protein in retinal ganglion cells - A possible sign of endogenous neuroprotection failure. Biochim Biophys Acta Mol Basis Dis. 2018 Jan;1864(1):296-306. doi: 10.1016/j.bbadis.2017.10.030. Epub 2017 Oct 28. PMID: 29107807.

  1. CD13 blot should be replaced. Fig 2D results (blot and quantification) are not convincing. The Y-axis label in Relative expression graph in Fig 2D is missing and the X-axis labeling should be consistent with other graphs.

We thank the reviewer for this comment. It is possible that due to the small amount of retinal tissue (10mg) and the lack of abundance of CD13 protein the Western blot may not be the adequate tool to evaluate the changes in the CD13 expression. We, instead opted, for the immunohistochemical analyses of blood vessels coverage with CD13+ pericytes, as was previously reported (Macura et al., 2022). We stained retinal sections (n=4 per group) with anti-CD13 (marker for pericytes) and Lectin (labels endothelial cells i.e., blood vessels). The % coverage was calculated as a ratio between ImageJ measurements in both channels. The results are presented in the graph (Figure 3.B).

Figure 3. The pericyte coverage of retinal blood vessels is decreased in 4 M 5xFAD, 12 M WT, and 12 M 5xFAD mice. (A) Representative images of the degree of pericyte coverage (CD13, red) of retinal blood vessels (Lectin, green) in the 4 M WT, 4 M 5xFAD, 12 M WT, and 12 M retinas. (B) The quantification of the CD13 retinal blood vessel coverage was performed using ImageJ and the results were presented as the ratios of CD13/Lectin staining (graph). Scale bar - 50mm.The data represent mean ± SEM value. ****p<0.0001.

  1. Figure 4. For Aqp4, the relative expression graph does not match the western blot because of which the results are not convincing. In 4C, Aquaporin 4 intensity should be quantified and better representative images should be included. Add scale bar.

We have eliminated the Aqp4 relative expression graph and the accompanying western blot. However, in order to strengthen the validity of results we added the additional experiment measuring the Aqp4 expression using immunohistochemistry (n=4 per group, Figure 5B). The Aqp4 expression was quantified using ImageJ. Statistical analysis is showed in the graph (Figure 5C).

              Figure 5. The expression levels of Aqp4 mRNA and protein were increased in the retinas from 4M and 12M 5xFAD and 12M WT mice. (A)The changes in the expression levels of Aqp4 mRNA. (B) The representative immunostaining of Aqp4 in the retinas of 4 M and 12 M, WT and 5xFAD mice. Aqp4 - red, DAPI – blue. (C) The changes in the protein expression levels of Aqp4 measured with ImageJ analysis. Scale bar – 50mmThe data represent mean ± SEM value. * p<0.05, ** p<0.01, *** p<0.001, **** p<0.0001. ONL – outer nuclear layer, INL – inner nuclear layer.

  1. Figure 5. Labels (A and B) and scale bars are missing. It appears from the images that 4-month old 5xFAD mice has reduced Aquaporin levels while the graph indicates no change in the levels. Please use a better representative image.

We have corrected Figure 5 (now Figure 6) according to the reviewer’s suggestions.

  1. It is recommended that the authors include a schematic diagram to summarize how Mfsd2a, Aqp4 etc are regulated.

We have added a schematic diagram explaining the potential mechanisms of Mfsd2a and Aqp4 actions.

Figure 7. Schematic representations of potential mechanisms of Mfsd2a and Aqp4 actions at the BRB in AD and during aging. (A) In the normally functioning BRB, Mfsd2a-mediated DHA transport inhibits the formation of caveolae possibly thorugh the inhibition of srebp1-c. In AD and aging Mfsd2a downregulation, associated with the loss of pericytes, lessens the inhibition of the formation of caveolae leading to the increased vesicle transport through BRB, leaky barrier and Ab accumulation. In AD the downregulation of Lxrb and Hmgcr could potentiate Mfsd2a downregulation. (B) Aqp4 expression is polarized, as it is densely expressed by astrocytes almost exclusively at the endfeet, in direct contact with the perivascular space. In aging and in AD Aap4 polarization is diminished. The loss of Aqp4 polarization is associated with the increased Ab accumulation.

  1. It is reported that in 5xFAD mice, Abeta accumulation starts as early as 1.5 months of age preceding amyloid deposition, gliosis and mitochondrial dysfunction at 2 months (Welding et al., 2019; Sharma et al., 2023). In this manuscript, the authors are trying to correlate Abeta clearance with Mfsda and Aqp4 levels. It is therefore crucial to include data from an earlier time point (preferable 2 months) to see if retinal accumulation of Abeta and changes in Mfsda and Aqp4 levels are coincidental. 

We thank the reviewer for this comment.

In this study we aimed to evaluate the changes in Mfsd2a and Aqp4 expression as putative targets for the development of new therapies that could improve vascular Ab clearance. 5xFAD mice are characterized with mild cognitive impairments as early as 4 months of age (Oakley et al., 2006). Considering that the diagnosis of AD before the first onset of clinical symptoms is still not possible, we chose to evaluate the changes in the Mfsd2a and Aqp4 expression at two time points - when the first signs of cognitive impairment present themselves (4 M) and when the pathology becomes more severe (12 M). Therefore, the focus of this study was primarily to evaluate the changes in Mfsd2a and Aqp4 expression in the retinas when the symptoms of AD pathology are already present. We agree with the reviewer that the logical next step would be to establish the time-line in order to correlate the retinal Mfsd2a and Aqp4 expression changes with cognitive changes and associate them with the Ab vascular accumulation. Furthermore, these changes should be than correlated with the similar changes in the brain. Most importantly, establishing the time-line can open the possibilities for the evaluation of the prophylactic potential of these molecules. This line of investigation is certainly part of our future research plans.

Oakley, H.; Cole, S.L.; Logan, S.; Maus, E.; Shao, P.; Craft, J.; Guillozet-Bongaarts, A.; Ohno, M.; Disterhoft, J.; Van Eldik, L.; et al. Intraneuronal β-Amyloid Aggregates, Neurodegeneration, and Neuron Loss in Transgenic Mice with Five Familial Alzheimer’s Disease Mutations: Potential Factors in Amyloid Plaque Formation. J. Neurosci. 2006, 26, 10129–10140. doi: 10.1523/JNEUROSCI.1202-06.2006

Reviewer 2 Report

In this manuscript, author investigated the changes in the expression levels of Mfsd2a and Aqp4 as regulators of Ab clearance from the blood vessels. Most of results were interesting to understand the impaired expression of Mfsd2a and Aqp4 in the retinas of 5xFAD mice underscoring their potential as targets for the development of new treatments aiming to improve the clearance of Ab from the blood vessels. The data are comprehensive and well presented in the figures, and the experimental approaches appear sound. However, the manuscript needs minor modifications to be considered by International Journal of Molecular Sciences.

Minor comments:

1)      Abstracts should be clearly represented Background, Purpose, Methods, Results, and Conclusions.  

2)      Introduction should provide scientific and logical backgrounds and reasons for research. It does not allow the description of simple concepts or action mechanisms to help readers understand.

3)      All subtitles in Materials and methods, Results and discussion, and Figure legends should be corrected. They did not represent some scientific means, but only methodological means.

4)      The name of each group was maintained the same form in all text and figure.

5)      Figure legends should be corrected to include title, information for figure, and information for abbreviation. Title should be changed from sentence to phrase.

6)      Figure 2D, Y axis should be explained.

7)      Table 1 should be included primer direction (5’ or 3’).

8)      Author should add the limitation of this study and further study in Discussion section.

9)      All number should be separated unit except % and oC. Also, unit should be described same pattern.

10)  Also, all abbreviation should be fully described when it firstly appeared. Also, this description should be not repeated in text.

11)  The information for all products, reagents and machines used this study should be clearly described based on the guide of journal. Ex) Product name (Company, Region, Country).

12)  Figure legends should be corrected to include title, information for figure, and information for abbreviation.

13)  References should be corrected according to journal guideline.

Author Response

Reviewer 2.

In this manuscript, author investigated the changes in the expression levels of Mfsd2a and Aqp4 as regulators of Ab clearance from the blood vessels. Most of results were interesting to understand the impaired expression of Mfsd2a and Aqp4 in the retinas of 5xFAD mice underscoring their potential as targets for the development of new treatments aiming to improve the clearance of Ab from the blood vessels. The data are comprehensive and well presented in the figures, and the experimental approaches appear sound. However, the manuscript needs minor modifications to be considered by International Journal of Molecular Sciences.

 Minor comments:

  • Abstracts should be clearly represented Background, Purpose, Methods, Results, and Conclusions.

We thank the reviewer for this comment. We have changed the Abstract according to the reviewer’s suggestions.

Abstract

Background Cerebral amyloid angiopathy (CAA) is characterized by amyloid b (Ab) accumulation in the blood vessels and is associated with the cognitive impairment in Alzheimer’s disease (AD).  The increased accumulation of Ab is also present in the retinal blood vessels and a significant correlation between retinal and brain amyloid deposition was demonstrated in living patients and animal AD models. Aim The Ab accumulation in the retinal blood vessels can be the result of impaired transcytosis and/or the dysfunctional ocular glymphatic system in AD and during aging. Methods We analyzed the changes in the mRNA and protein expression of major facilitator superfamily domain-containing protein2a (Mfsd2a), the major regulator of transcytosis, and of Aquaporin4 (Aqp4), the key player implicated in the functioning of the glymphatic system, in the retinas of 4-and 12-month-old WT and 5xFAD female mice. Results A strong decrease in the Mfsd2a mRNA and protein expression was observed in the 4 M and 12 M 5xFAD and 12M WT retinas. The increase in the expression of Srebp1-c could be at least partially responsible for the Mfsd2a decrease in the 4 M 5xFAD retinas. The decrease in the pericyte (CD13+) coverage of retinal blood vessels in the 4 M and 12 M 5xFAD retinas and in the 12 M WT retinas suggests that pericyte loss could be associated with the Mfsd2a downregulation in these experimental groups. The observed increase in Aqp4 expression in 4 M and 12 M 5xFAD and 12 M WT retinas accompanied by the decreased perivascular Aqp4 expression is indicative of the impaired glymphatic system. Conclusion The findings in this study reveal the impaired Mfsd2a and Aqp4 expression and Aqp4 perivascular mislocalization in retinal blood vessels during physiological (WT) and pathological (5xFAD) aging indicating their importance as putative targets for the development of new treatments that can improve the regulation of transcytosis or the function of glymphatic system.

  • Introduction should provide scientific and logical backgrounds and reasons for research. It does not allow the description of simple concepts or action mechanisms to help readers understand.

We have changed the introduction according to the reviewer’s suggestions. All the changes are highlighted yellow in the revised version of the manuscript.

3)      All subtitles in Materials and methods, Results and discussion, and Figure legends should be corrected. They did not represent some scientific means, but only methodological means.

We have changed the subtitles in the MS according to the reviewer’s suggestions.

4)      The name of each group was maintained the same form in all text and figure.

We have introduced the adequate corrections.

5)      Figure legends should be corrected to include title, information for figure, and information for abbreviation. Title should be changed from sentence to phrase.

We have incorporated the changes suggested by the reviewer to all figure legends.

6)      Figure 2D, Y axis should be explained.

We have incorporated adequate changes.

7)      Table 1 should be included primer direction (5’ or 3’).

We have included the primer direction in Table 1.

8)      Author should add the limitation of this study and further study in Discussion section.

We thank the reviewer for this comment.

This study was performed on the retinas from 5xFAD AD mouse model. 5xFAD mice are considered an aggressive model for Alzheimer’s disease expressing human amyloid precursor protein (APP) and presenilin 1 (PSEN1) transgenes with a total of five AD-linked mutations (three in APP and two in PSEN1) under transcriptional control of neuron-specific murine Thy-1 promoter (Oakley et al., 2006). The extremely aggressive Aβ pathology, in 5xFAD mice with extensive extracellular plaque formation beginning at 2 months of age, is much faster and more hostile compared to the time course of human sporadic AD.Therefore, it will be beneficial to examine Mfsd2a and Aqp4 expression changes in more humanized AD mouse models such as APP knock-in for example (Xia et al., 2022). Considering that a significant correlation between retinal and brain amyloid deposition was observed in AD patients and animal models further studies focusing on examining if there is a correlation between the changes in Mfsd2a and Aqp4 expression in brain versus retinal blood vessels are warranted.

Oakley, H.; Cole, S.L.; Logan, S.; Maus, E.; Shao, P.; Craft, J.; Guillozet-Bongaarts, A.; Ohno, M.; Disterhoft, J.; Van Eldik, L.; et al. Intraneuronal β-Amyloid Aggregates, Neurodegeneration, and Neuron Loss in Transgenic Mice with Five Familial Alzheimer’s Disease Mutations: Potential Factors in Amyloid Plaque Formation. J. Neurosci. 2006, 26, 10129–10140. doi: 10.1523/JNEUROSCI.1202-06.2006

Xia D, Lianoglou S, Sandmann T, Calvert M, Suh JH, Thomsen E, Dugas J, Pizzo ME, DeVos SL, Earr TK, Lin CC, Davis S, Ha C, Leung AW, Nguyen H, Chau R, Yulyaningsih E, Lopez I, Solanoy H, Masoud ST, Liang CC, Lin K, Astarita G, Khoury N, Zuchero JY, Thorne RG, Shen K, Miller S, Palop JJ, Garceau D, Sasner M, Whitesell JD, Harris JA, Hummel S, Gnörich J, Wind K, Kunze L, Zatcepin A, Brendel M, Willem M, Haass C, Barnett D, Zimmer TS, Orr AG, Scearce-Levie K, Lewcock JW, Di Paolo G, Sanchez PE. Novel App knock-in mouse model shows key features of amyloid pathology and reveals profound metabolic dysregulation of microglia. Mol Neurodegener. 2022 Jun 11;17(1):41. doi: 10.1186/s13024-022-00547-7. PMID: 35690868; PMCID: PMC9188195.

9)      All number should be separated unit except % and oC. Also, unit should be described same pattern.

We have corrected all the measurement units in the MS.

10)  Also, all abbreviation should be fully described when it firstly appeared. Also, this description should be not repeated in text.

We have corrected all the abbreviations in the text and eliminated all the repetitions in the MS.

11)  The information for all products, reagents and machines used this study should be clearly described based on the guide of journal. Ex) Product name (Company, Region, Country).

We have provided the complete manufacturer’s information for all products, reagents, and machines used.

12)  Figure legends should be corrected to include title, information for figure, and information for abbreviation.

Please see the response for the point 5.

13)  References should be corrected according to journal guideline.

We have organized references according to the journal guidelines.

Round 2

Reviewer 1 Report

The authors have done a good job with their response to the Reviewers's comments. The manuscript has improved immensely and the results justify the conclusions. 

There are still some minor issues with the English language which can be improved (grammar).